# Tick-Borne Pathogens Shape the Native Microbiome Within Tick Vectors

**DOI:** 10.3390/microorganisms8091299

**Published:** 2020-08-25

**Authors:** Abdulsalam Adegoke, Deepak Kumar, Cailyn Bobo, Muhammad Imran Rashid, Aneela Zameer Durrani, Muhammad Sohail Sajid, Shahid Karim

**Affiliations:** 1Center for Molecular and Cellular Biosciences, School of Biological, Environmental and Earth Sciences, University of Southern Mississippi, Hattiesburg, MS 39406, USA; Abdulsalam.Adegoke@usm.edu (A.A.); Deepak.Kumar@usm.edu (D.K.); Cailyn.Bobo@usm.edu (C.B.); 2Department of Parasitology, Faculty of Veterinary Science, The University of Veterinary and Animal Sciences, Lahore 54000, Pakistan; imran.rashid@uvas.edu.pk; 3Department of Clinical Medicine and Surgery, Faculty of Veterinary Science, The University of Veterinary and Animal Sciences, Lahore 54000, Pakistan; aneela@uvas.edu.pk; 4Department of Parasitology, Faculty of Veterinary Science, University of Agriculture, Faisalabad 38000, Pakistan; drsohailuaf@hotmail.com

**Keywords:** ticks, microbiome, *Hyalomma anatolicum*, *Rhipicephalus microplus*, *Anaplasma marginale*, *Theileria* sp., *Francisella*, *Wolbachia*, Pakistan

## Abstract

Ticks are blood-feeding arthropods and transmit a variety of medically important viral, bacterial, protozoan pathogens to animals and humans. Ticks also harbor a diverse community of microbes linked to their biological processes, such as hematophagy, and hence affect vector competence. The interactions between bacterial and/or protozoan pathogens and the tick microbiome is a black-box, and therefore we tested the hypothesis that the presence of a protozoan or bacterial pathogen will alter the microbial composition within a tick. Hence, this study was designed to define the microbial composition of two tick species, *Hyalomma (H.) anatolicum* and *Rhipicephalus (R.) microplus*. We used a combination of PCR based pathogen (*Anaplasma marginale* and *Theileria* species) and symbiont (*Wolbachia* species) identification followed by metagenomic sequencing and comparison of the microbial communities in PCR positive and negative ticks. A total of 1786 operational taxonomic units was identified representing 25 phyla, 50 classes, and 342 genera. The phylum Proteobacteria, Firmicutes, Actinobacteriota, and Bacteroidota were the most represented bacteria group. Alpha and beta diversity were not significantly affected in the presence or absence of *Theileria* sp. and *A. marginale* as see with *H. anatolicum* ticks. Interestingly, bacterial communities were significantly reduced in *Theileria* sp. infected *R. microplus* ticks, while also exhibiting a significant reduction in microbial richness and evenness. Putting these observations together, we referred to the effect the presence of *Theileria* sp. has on *R. microplus* a “pathogen-induced dysbiosis”. We also identify the presence of *Plasmodium falciparum*, the causative agent of human malaria from the microbiome of both *H. anatolicum* and *R. microplus* ticks. These findings support the presence of a “pathogen-induced dysbiosis” within the tick and further validation experiments are required to investigate how they are important in the vector competence of ticks. Understanding the mechanism of “pathogen-induced dysbiosis” on tick microbial composition may aid the discovery of intervention strategies for the control of emerging tick-borne infections.

## 1. Introduction

Ticks are obligate, blood-feeding ectoparasites of vertebrate animals that depend on the host’s blood for nutrition and reproduction. They elicit significant blood loss and also transmit disease-causing bacteria, viruses, and protozoa from one host to another, which makes them significant to public health. In Pakistan, livestock farming and production serve as one of the major drivers of the macro-economy as with the latest animal census showing an estimated population of 40 million, 47.8 million, 76.1 million, and 30.9 million buffalo, cattle, goats, and sheep, respectively, all contributing to an estimated 11.22% of the country’s gross domestic product (GDP) [1].

Previously reported tick species infesting livestock in Pakistan includes both hard and soft tick species. In Pakistan, the most reported ticks found infesting livestock are hard ticks of the genus Haemaphysalis, Hyalomma, and Rhipicephalus, and soft ticks of the genus Argas and Ornithodoros [2,3,4,5]. Hyalomma and Rhipicephalus ticks are responsible for transmitting major tick-borne pathogens that affect livestock animals in Pakistan. *Anaplasma marginale, Anaplasma centrale*, *Babesia bovis,* and *Babesia bigemina* are which are known to cause cattle fever are transmitted by *Rhipicephalus microplus* ticks [6]. Tropical theileriosis, a small ruminant disease caused by *Theileria annulata* and the Crimean Congo hemorrhagic fever virus have been reported to be transmitted by *Hyalomma anatolicum* [6,7].

Ticks also harbor several distinct microbial communities, members of which have been shown to play an important role in tick biology. Analysis of the genome of such microbes has revealed specific regions coding for essential vitamins, most of which are lacking in the tick’s blood meal [8,9,10], emphasizing their possible role as nutritional mutualists. Recent evidence also suggests that some members of these microbial communities can potentially interact with tick-borne pathogens both directly and indirectly. It was reported that removing the midgut bacteria of black-legged ticks *Ixodes scapularis* by feeding on antibiotic-treated mice impairs infection by *Borrelia burgdorferi* [11]. Another tick-borne pathogen, *Anaplasma. phagocytophilum* was shown to reduce the viability of the microbial population, thus facilitating its colonization of the *I. scapularis* midgut [12].

Along with the development in tick microbiome studies, adequate information exists on how tick-borne human pathogens shape the tick microbial communities to facilitate their colonization and subsequent transmission. While adequate research has been carried out on the microbiome of livestock infesting ticks, there is very little scientific understanding of the interactions that occurs between tick-transmitted pathogens and the tick microbial communities. Apart from Karim et al. [4], who reported important bacterial genera found in ticks from Pakistan, there is a general lack of research in understanding how specific bacterial or protozoan pathogens of animal origin shapes the microbiome of several tick vectors.

The main purpose of this study is to develop an understanding of changes that occurs in the microbial composition within a tick vector when such a tick acquires a bacterial or protozoan pathogen of animal source. The key research question of this study was whether or not the overall abundance and diversity of tick’s microbial communities are reduced in the presence of a protozoan or bacteria pathogen. This study provides an exciting opportunity to advance our knowledge of tick microbiome and tick-borne pathogen interactions. A full discussion of the molecular mechanisms of reported interactions lies beyond the scope of this study. We will proceed to investigate specific microbial changes and interactions as part of our ongoing study. It is also beyond the scope of this study to examine whether host blood meal impacts microbial diversity.

## 2. Materials and Methods

### 2.1. Tick Collection and Identification

Fully engorged adult *Hyalomma anatolicum* and *Rhipicephalus microplus* ticks were randomly and carefully removed from livestock animals from four livestock producing regions in Pakistan (Sialkot [32°29′33.7″N, 74°31′52.8″E], Gujrat [32°34′22″N, 74°04′44″ E], Gujranwala [32°9′24″N, 74°11′24″E], and Sheikhupura [31°42 47″ N, 73°58′41″ E]). This was done by careful removal of fully engorged ticks using tweezers with care been taken to keep the mouthparts intact. All ticks were kept in separate vials containing 70% ethanol and details of the location, and the host was recorded. For this study, a total of 320 ticks were selected and shipped from Pakistan to the University of Southern Mississippi for further analysis using the U.S. Department of Agriculture’s Animal and Plant Health Inspection Service (permit # 11122050). Identification of ticks to the genus level was carried out by an expert taxonomist at the United States National Tick Collection (USNTC) according to the criteria used in previously published reports [2,13,14]. All stages were examined on an Olympus SZX16 stereoscopic microscope (Olympus Life Science, Center Valley, PA, USA). To further confirm morphological identification, ticks homogenates were subjected to molecular identification by amplifying the highly conserved 708 bp mitochondrial Cytochrome Oxidase I gene (COI) [15]. The nucleotide accession numbers were MT876643, MT876644, and MT876645.

### 2.2. Genomic DNA Extraction

High-quality DNA was extracted from all the 320 ticks. Before DNA extraction, ticks were removed from the transport vials, cleaned using 100% ethanol, dried, and subsequently cleaned using a 10% sodium hypochlorite solution. Ticks were finally cleaned using distilled water and allowed to dry on a kimwipe paper. Homogenization of individual ticks was done mechanically, first by cutting ticks into smaller pieces, followed by complete disruption using an automated, hand-held homogenizer. The DNeasy Blood & Tissue Kit (Qiagen, Germantown, MD, USA) was used to extract DNA from individual ticks with minor modification in the volume eluted (30 µL). The DNA concentrations and quality were quantified using a nanodrop machine (Nanodrop One, Thermo Fisher Scientific, Pittsburgh, PA, USA) and DNA stored in −20 °C till further needed.

### 2.3. Detection of Pathogen and Endosymbiont

To detect the presence of pathogens and endosymbiont of interest, we utilized a PCR based approach to amplify the 18S rRNA gene of *Theileria* sp. [4], 16S rRNA gene of *Anaplasma marginale* [16], and GroEL gene of Wolbachia [17]. PCR positive DNAs were amplicon sequenced using both the forward and reverse primers and the partial sequences were subjected to NCBI BLAST program for further confirmation. Details of the primers, conditions, and amplicon sizes can be found in Table 1. The *Wolbachia* sp. nucleotide accession numbers are MT881679, MT881680, MT881681, MT881682, MT881683, MT881684, MT881685, MT881686, MT881687, MT881688, MT881689, MT881690, MT881691, MT881692, MT881693, MT881694, MT881695, MT881696, MT881697, MT881698, MT881699, MT881700, MT881701 and MT881702.

### 2.4. 16S rRNA Library Preparation and Sequencing

A total of 40 ticks (20 *H. anatolicum* and 20 *R. microplus*) were used for microbiome analysis. From each tick species, 5 *Theileria* sp. positive, 5 *A. marginale* positive, and 10 negative ticks were selected for 16S rRNA library preparation and sequencing. The hypervariable V1–V3 region of the 16S rRNA gene was PCR amplified using the forward primer 27F (5’-AGR GTT TGA TCM TGG CTC AG-3’) and the reverse primer 519R (5’-GTN TTA CNG CGG CKG CTG-3’) as outlined by the 16S Illumina’s MiSeq protocol (www.mrdnalab.com, Shallowater, TX, USA. Accessed on 11 July 2020). Briefly, PCR was performed using the HotStarTaq Plus Master Mix Kit (Qiagen, Germantown, Maryland, USA) under the following conditions: 94 °C for 3 min, followed by 30–35 cycles of 94 °C for 30 s, 53 °C for 40 s and 72 °C for 1 min, after which a final elongation step at 72 °C for 5 min was performed. After amplification, PCR products were electrophoresed in 2% agarose gel to determine the success of amplification and the relative intensity of bands. Multiple samples were pooled together in equal proportions based on their molecular weight and DNA concentrations. Pooled samples were purified using calibrated Ampure XP beads. Then the pooled and purified PCR product was used to prepare Illumina DNA library. Sequencing was performed at MR DNA (www.mrdnalab.com, Shallowater, TX, USA. Accessed on 11 July 2020) on a MiSeq following the manufacturer’s guidelines.

### 2.5. Sequence Analysis

Sequence analysis was carried out using the Quantitative Insights into Microbial Ecology (QIIME 2) pipeline unless stated otherwise. Briefly, the processing of raw fastq files was demultiplexed. The Atacama soil microbiome pipeline was incorporated for quality control of demultiplexed paired-end reads using the DADA2 plugin as previously described [18]. Low-quality sequences were trimmed and filtered out, and subsequent merging of paired-end-reads was done ensuring 20 nucleotide overhang between forward and reverse reads. Chimeric sequences were removed from the sequence table.

Sequence alignment and subsequent construction of phylogenetic tree from representative sequences were performed using the MAFFT v. 7 and FasTree v. 2.1 plugins [19] Operational taxonomic assignment was performed using the qiime2 feature-classifier plugin v. 7.0, which was previously trained against the SILVA 138 database preclustered at 99%. Tables representing operational taxonomic units (OTUs) and representative taxonomy were exported from R and used for diversity metric analysis using the Microbiome Analyst web-based interface [20,21]. Raw data from this analysis were submitted deposited and assigned the GenBank BioProject number #PRJNA600935.

### 2.6. Alpha Diversity

To establish whether alpha diversity differs across tick samples, reads were transformed and low abundance OTUs were filtered from the datasets. The Observed OTU metric was used to estimate species richness by identifying unique OTUs present across the tick groups, while the Shannon index was used to estimate both richness and evenness.

### 2.7. Beta Diversity

To compare the differences in the microbiome between tick groups, based on measures of distance or dissimilarity, dissimilarity matrix was generated from log-transformed sequence data and ordination of the plots was visualized using both the Principal Coordinates Analysis (PCoA) and the Nonmetric Multidimensional Scaling (NMDS). The Bray–Curtis distance matrix was used to visualize compositional differences in the microbiome across all groups.

### 2.8. Statistical Analysis

Statistical significance was inferred using the Mann–Whitney/Kruskal–Wallis method for alpha diversity and classical univariate comparison analysis, while the Permutational MANOVA (PERMANOVA) was used to test for the statistical significance of the dissimilarity measures.

## 3. Results

### 3.1. Pathogen and Symbiont Prevalence

PCR analysis supported by amplicon sequencing and blast analysis of sequenced PCR product showed that 23 (7.2%), 90 (28.1%), and 3 (0.9%) of the 320 individually tested ticks were positive for *A. marginale*, *Theileria* sp., and *Wolbachia* sp. (Appendix A). We further determined the genetic relationship of the identified *Wolbachia* sp. and the tick species used with publicly available sequences from NCBI (Appendix A). The Wolbachia GroEL gene identified in ticks from this study shows high similarity to *Wolbachia pipientis* strain wAlbB-HN2016 and wAlbB-FL2016, with 99% query cover and 99.61% identity (Appendix A). The query cover and percentage identities of the COI sequences from this study were also compared to those previously deposited in the NCBI database (Appendix A).

### 3.2. Bacteria 16S rRNA Abundance Profile

A total of 2,787,815 million reads paired-end reads were generated. Analysis of the demultiplexed paired-end-reads generated 2,787,815 reads which ranged from 36,124 to 123,736 with an average of 65,609 reads. After passing the sequences through quality filtering, 119,802 of the raw reads were non-chimeric which were subsequently used for taxonomic classification (Appendix A). Taxonomic classification using the SILVA reference base identified 472 OTUs generated from *R. microplus* ticks belonging to 10 phyla, 17 classes, and 146 genera. *H. anatolicum* had a total of 1314 OTUs representing 15 phyla, 33 classes, and 196 genera.

### 3.3. Bacteria Relative Abundance

The relative proportion of bacteria at different taxonomic classification was further analyzed in both tick species. Figure 1 presents the results obtained from the taxonomic classification of identified bacteria OTU at phylum, family, and genus taxonomic levels. Additional figures showing relative abundances of bacteria species in individual samples can be found (Appendix A).

The phylum Proteobacteria, Firmicutes, Actinobacteriota, and Bacteroidota were all found to be present in *H. anatolicum* ticks. As shown in Figure 1A, the phylum Proteobacteria was found to be present at an abundance of 87.5%, 68%, and 49% in *Theileria* sp. positive, uninfected and *A. marginale* positive *H. anatolicum* ticks, respectively, while Firmicutes (25%) was only present in *A. marginale* infected ticks. Phylum level abundance in *R. microplus* ticks (Figure 1B) contrasts that shown in *H. anatolicum*. The entirety of the bacteria identified in *Theileria* sp. positive *A. marginale* belongs to the phylum Firmicutes (100%), while both *A. marginale* positive and positive *R. microplus* shares similar bacteria phylum distribution representing Actinobacteria, Bacteroidota, and Proteobacteria (Figure 1B).

Francisellaceae (37.5%) and Rickettsiales_*fa* (50%) constituted to approximately 87.5% of the bacteria family identified within *Theileria* sp. positive *H. anatolicum* ticks. These bacteria families were also identified in uninfected ticks albeit at a much-reduced abundance (Figure 1C). Staphylococcaceae was identified at a relative abundance of ~30% in *A. marginale* infected *H. anatolicum*. The family Anaplasmataceae (37.5%) were identified at similar abundance in both *A. marginale* infected and uninfected *H. anatolicum* (Figure 1C). Distribution of the bacteria family in *R. microplus* ticks identified Bacillaceae at a 100% abundance in *Theileria* sp. positive *R. microplus*, while the family Corynebacteriaceae, Coxiellaceae, Flavobacteriaceae, Staphylococcaceae, and *Weeksellaceae* were detected in similar abundances in *A. marginale* positive and uninfected *R. microplus* (Figure 1D).

Similar differences in the bacteria abundances were further identified at the genus level which reflects those seen in the family and phylum. As can be seen from Figure 1E, five major genera; *Acinetobacter*, *Anaplasma*, *Devosia*, *Norcadiopsis*, and *Sphingomonas* which represents 87.5% of the bacteria were identified from *A. marginale* infected *H. anatolicum* ticks. The only genus of bacteria identified in *Theileria* sp. positive *H. anatolicum* ticks was *Candidatus*_Midichloria (51.5%) and *Francisella* (36%). The genus *Candidatus*_Midichloria and *Francisella* in addition to *Ehrlichia*, *Hydrobacter*, and *Corynebacterium* were identified in uninfected ticks (Figure 1E). Figure 1F shows similar bacterial composition at the genus level between uninfected and *A. marginale* positive *R. microplus* ticks, while the only identified genus in *Theileria* sp. positive *R. microplus* is Bacillus. These results of the bacteria abundance indicate that there is an association between the presences of *Theileria* sp. and how it shapes the bacteria composition of the two different tick species.

### 3.4. Eukaryote 18S rRNA Abundance

We equally identified and compared eukaryote species in both ticks. Surprisingly, we detected the presence of *Plasmodium falciparum* in both tick groups, and *Hepatozoon americanum* in *H. anatolicum* ticks, both of which have a higher abundance in the *Theileria* sp. positive ticks (Figure 2A,B).

### 3.5. Microbial Richness and Evenness

Microbial profile richness and evenness were estimated using the alpha diversity metrics observed OTUs and Shannon index. There was no significant difference between richness and evenness within the *H. anatolicum* ticks, irrespective of the PCR status (Appendix A). Interestingly, *R. microplus* ticks showed comparable significant differences between *Theileria* sp. positive and uninfected ticks (Figure 3A,B). *R. microplus* ticks positive for *Theileria* sp. exhibited significantly reduced species richness and evenness index across all metrics used to analyze alpha diversity.

### 3.6. Microbial Similarity/Dissimilarity Patterns

Differences in the microbial communities were analyzed using Bray–Curtis and Jaccard distance matrices. No significant observation was made in beta diversity across the *H. anatolicum* ticks (Appendix A). Beta diversity was significantly different in *R. microplus* by both Bray–Curtis (PERMANOVA, F-value: 4.2171; R-squared: 0.33161; *p*-value < 0.005; Figure 4A) and Jaccard (PERMANOVA, F-value: 3.2588; R-squared: 0.27714; *p*-value < 0.005; Figure 4B). Non-metric multidimensional scaling (NMDS) plot of microbial communities further showed a distinct separation of *Theileria* sp. positive *R. microplus* ticks from both *A. marginale* and uninfected (Appendix A).

### 3.7. Community Profiling and Correlation Analysis of R. microplus Ticks

To assess the extent to which highly abundant bacteria phylum and genus were represented in *R. microplus* ticks, we used a combination of pattern correlation and heat map analysis. A very strong positive correlation was seen between the presence *of Bacillus* and *Theileria* sp. positive *R. microplus* ticks (Figure 5). A similar observation was also seen in the heat map where the genus *Bacillus* shows the highest presence in *Theileria* sp. positive *R. microplus* ticks (Figure 6A).

To explore how top taxa differ, classical univariate statistical comparisons analysis was applied to identify bacterial genus that exhibits significant differences (*t*-test/ANOVA) in their composition. Significant differences were observed in the abundance of the genus Acinetobacter, Staphylococcus, and Bacillus. Heat map analysis of OTU abundance was also estimated for *H. anatolicum* ticks, none of which was statistically significant (Appendix A).

In summary, the results of alpha and beta diversity as well as correlation analysis suggest that a strong association exists between *Theileria* sp., reduced alpha diversity metrics, and the distinct clustering separation exhibited by *Theileria* sp. positive *R. microplus* ticks.

## 4. Discussion

The present study was designed to determine the changes that occur to the microbiome composition and diversity within the tick vectors when naturally infected with protozoan and bacterial tick-borne pathogens. Although many studies have detailed the plethora of interactions that occur between tick-transmitted pathogens and the microbiome of ticks such as endosymbionts and pathogen interactions [22], pathogen induction of antimicrobial production by the tick vector [12,23], only one of such studies compared the microbiome of pathogen-infected and uninfected ticks [24]. In this study, we tested field-collected *H. anatolicum* and *R. microplus* ticks for the presence of *Theileria* sp., *Wolbachia* sp., and *A. marginale*, and further compared the overall microbial distribution, richness, and diversity between *Theileria* sp. and *R. microplus* positive ticks. In the current study, the estimated percentage of *Theileria* sp. positive ticks was higher compared to those reported in previous studies for *H. anatolicum*, while *A. marginale* prevalence was similar to previous reports [6,25,26].

The major bacteria phyla reported across all the tick groups irrespective of the PCR status were Proteobacteria, Bacteroidota, Firmicutes, and Actinobacteriota. These support observations from earlier tick microbiome studies [4,11,27,28]. Surprisingly, we identified a much lower number of bacterial reads and OTU from *R. microplus* ticks compared to *H. anatolicum*. This observation could be a function of the differences in the lifecycle of the two tick species. *Rhipicephalus microplus* is known as a one-host tick, whereas *H. anatolicum* ticks could spend their life cycle using 2–3 hosts. Spending different life stages on different animal hosts will likely expose the *H. anatolicum* ticks to a plethora of host skin microbial communities as well as host-blood [29] associated microbial communities. Some of the identified bacterial genera from *H. anatolicum* ticks such as Staphylococcus, Corynebacterium, Sphingomonas, and Cutibacterium have been previously shown to be common constituents of the skin microflora [29].

We also observed the presence of *Theileria* sp. was associated with an increased abundance of Candidatus_Midichloria and Francisella compared to the uninfected or A. marginale infected H. anatolicum. Candidatus_Midichloria and Francisella are obligate, vertically maintained endosymbionts in the phylum Proteobacteria. Candidatus_Midichloria belongs to the Alphaproteobacteria group of obligate intracellular bacteria first detected in *Ixodes ricinus* [30], while *Francisella*-like endosymbiont is a Gammaproteobacteria with widespread distribution in hard ticks [31]. It is interesting to compare these findings to an elegant observation made by Budachetri and colleagues [22] who proposed *Candidatus_Midichloria mitochondrii as* facilitating *Rickettsia parkeri* colonization of the *Amblyomma maculatum* tissues by protecting *R. parkeri* from the deleterious effect of reactive oxygen species. Our observations do require further experimental validation to understand the interaction between *Theileria* sp. and Candidatus_Midichloria within the tick vector.

The detection of *Bacillus* as the only bacteria genera identified from *Theileria* sp. positive *R. microplus* ticks was an unexpected, albeit important finding (Figure 1F). This observation was further validated by a significantly strong association between the presence of *Theileria* sp. and the *Bacillus* group of bacteria as seen in Figure 5 and Figure 6A. We also found a higher abundance of the phylum Firmicutes (Appendix A) and genus Bacillus was associated with *Theileria* sp. presence in *R. microplus* ticks (Figure 6). It is difficult to explain this result, but it might be related to the ability of *Theileria* sp. to interfere with the mammalian host’s immune response by expressing proteins necessary for its transformation [32]. However, its impact on the tick microbiota has yet to be shown. Inhibition of important microbial metabolic pathways by *Theileria* sp.-associated proteins [33] could have led to a pathogen-associated dysbiosis. Bacillus ability to form spores when exposed to unfavorable physiological conditions would explain their exclusive presence in *Theileria* sp. positive *R. microplus* ticks.

We also observed that *R. microplus* ticks exhibited a significantly lower microbial diversity and composition when compared to *H. anatolicum* ticks. Within the *R. microplus,* those positive with *Theileria* sp. showed a significantly lesser amount of identified bacteria OTUs and a significantly reduced species richness and evenness (Figure 3A,B), while also displaying a different microbial composition as observed on the ordination plots (Figure 4A,B; Appendix A). These observations were consistent with those of Mann et al. [34], who reported that *Trypanosoma cruzi* infected kissing bugs exhibited a higher abundance of selected bacteria group, but not consistent with a study by Swei and Kwan [29] who reported that *Ixodes pacificus* ticks positive with the Lyme disease spirochete, *Borrelia burgdorferi,* had no significant differences in the microbiome richness and composition when compared to those not infected.

Another interesting finding from our study was the detection of the vertically transmitted *Wolbachia* sp. bacteria from a small number of the ticks used in this study. Wolbachia is an endosymbiont that has been identified in two-thirds of insects, including mosquitos [35], with known ability to interfere with their host’s reproduction [36] through a series of physiological alterations, one of which is cytoplasmic incompatibility [37]. Studies showed the presence of Wolbachia in tick species has associated it with a form of hyperparasitism where Wolbachia was found infecting another parasitoid that parasitizes the ticks [38,39].

The unexpected identification of *Plasmodium falciparum* from ticks used in this study is an unpredicted finding. Pakistan is a malaria-endemic country and we observed an increase in the relative abundance of *Plasmodium falciparum* in *Theileria* sp. positive *H. anatolicum* when compared to the uninfected ticks (Figure 2A,B). It seems possible that this could have occurred from the ticks accidentally feeding on a *P. falciparum*-infected human host. While this is a possibility in multi-host ticks as seen in *Hyalomma* species, this is highly an unlikely occurrence in *R. microplus* which is a one-host tick. These results, therefore, need to be interpreted with caution as ticks are not reservoirs or competent vectors of *P. falciparum*.

This finding, while preliminary, suggests that the presence of *Theileria* sp. within *R. microplus* ticks reduces the overall microbial diversity which we proposed as “pathogen-induced dysbiosis”. The mechanism behind this phenomenon could be induced by *Theileria* sp. factors in an attempt to colonize the tick vector, or it could be a result of the innate immune response mounted by the tick. These findings may help us to understand the intricate interplay of the pathogen–microbiome–vector interactions. However, more research on these observed interactions needs to be undertaken before the association between microbiome dysbiosis and the presence of a pathogen can be drawn.

## 5. Conclusions

The present study was designed to determine the effect of bacteria and protozoan pathogen on the microbiome of field-collected ticks. Using a combination of PCR based assay and 16S rRNA sequencing, we investigated how the presence of *Theileria* sp. and *A. marginale* shapes the overall microbiome of both *H. anatolicum* and *R. microplus* ticks. We reported a strong association between the presence of *Theileria* sp. and a completely reduced microbial diversity and abundance in *R. microplus* ticks. This study established the extent of the diversity of the microbial community within two important tick species from Pakistan and revealed the presence of *Theileria* sp., *A. marginale,* and additional pathogenic bacteria that could be of public health significance. A limitation of this study was the difficulty in obtaining tissue samples of ticks, as they were field collected. Future tick developmental and tissue-specific studies will generate new insights into specific interactions between tick-borne pathogens and their associated microbiomes.

## Figures and Tables

**Figure 1 microorganisms-08-01299-f001:**
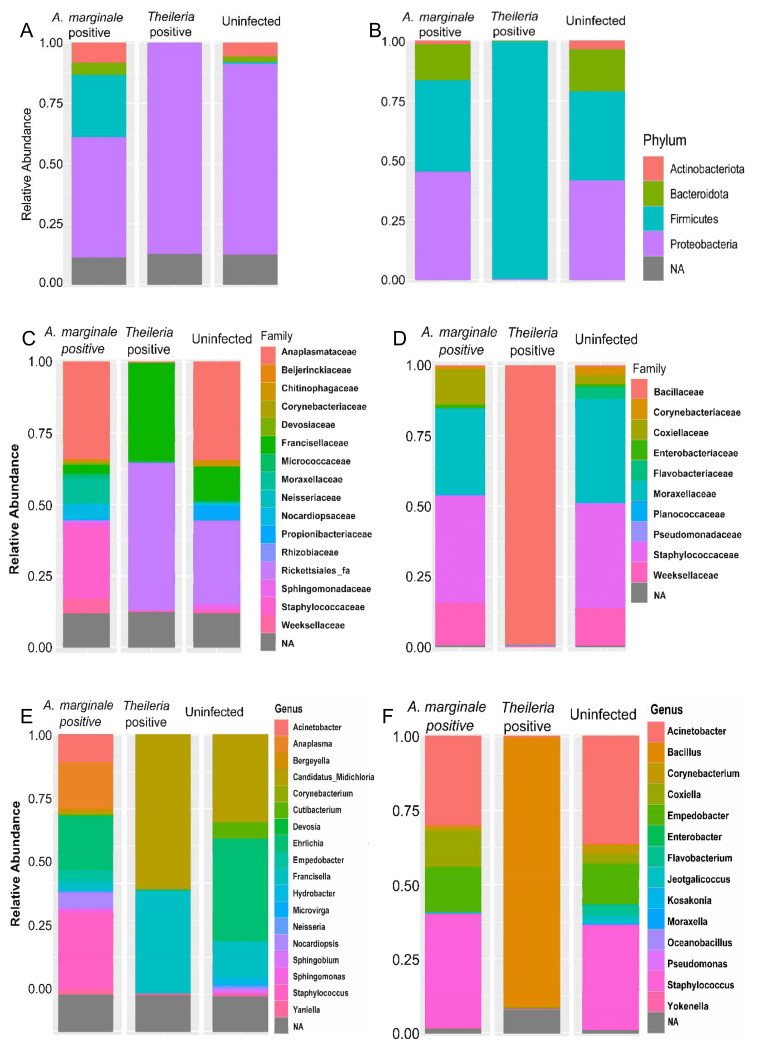
16S bacteria abundance profiles of *H. anatolicum* and *R. microplus*. (**A**,**B**) Bacteria abundance at phylum. (**C**,**D**). Bacteria abundance at the family level. (**E**,**F**) Bacteria abundance at the genus level. *Hyalomma anatolicum* (left panel), *R. microplus* (right panel).

**Figure 2 microorganisms-08-01299-f002:**
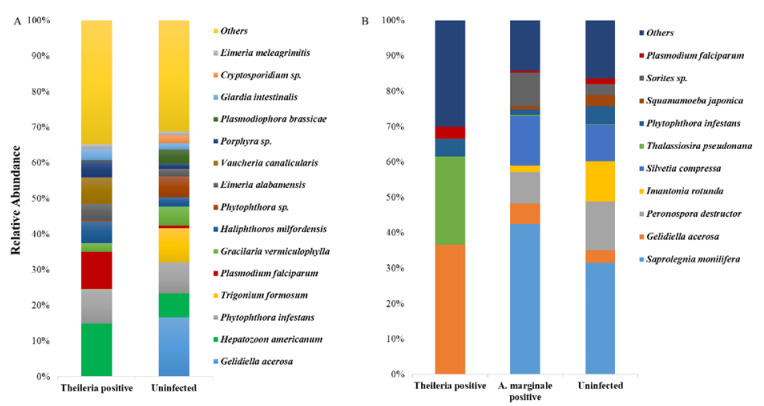
18S eukaryote abundance profiles of *H. anatolicum* and *R. microplus*. (**A**) The relative abundance of identified eukaryote species from *Theileria* sp. positive and uninfected H. anatolicum ticks. (**B**) The relative abundance of identified eukaryote species from *Theileria* sp. positive, *A. marginale* positive, and uninfected *R. microplus* ticks.

**Figure 3 microorganisms-08-01299-f003:**
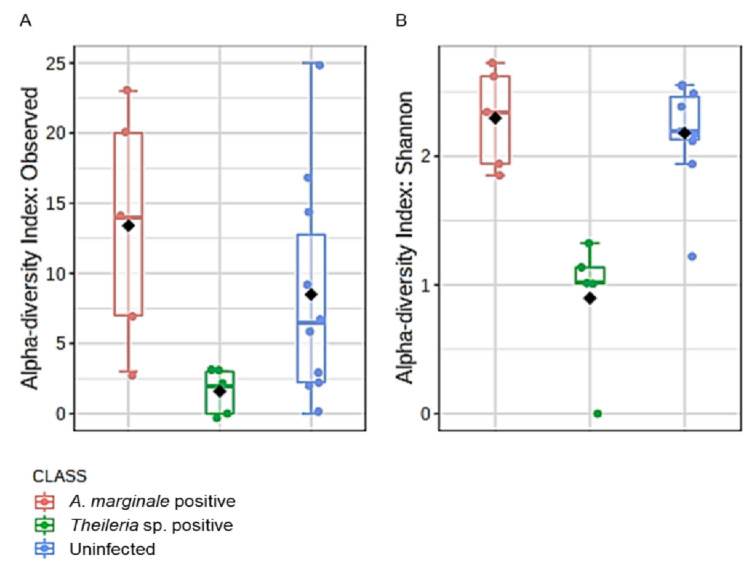
Alpha diversity analysis in *R. microplus* ticks. (**A**) Estimation of species richness using the observed operational taxonomic (OTUs) metrics (Kruskal–Wallis H-test, df = 1, *p*-value = 0.003). (**B**) Estimation of species evenness using the Shannon diversity index (Kruskal–Wallis H-test, df = 1, *p*-value = 0.005). Ticks found to be positive for *Theileria* sp. showed the least diversity using both measures.

**Figure 4 microorganisms-08-01299-f004:**
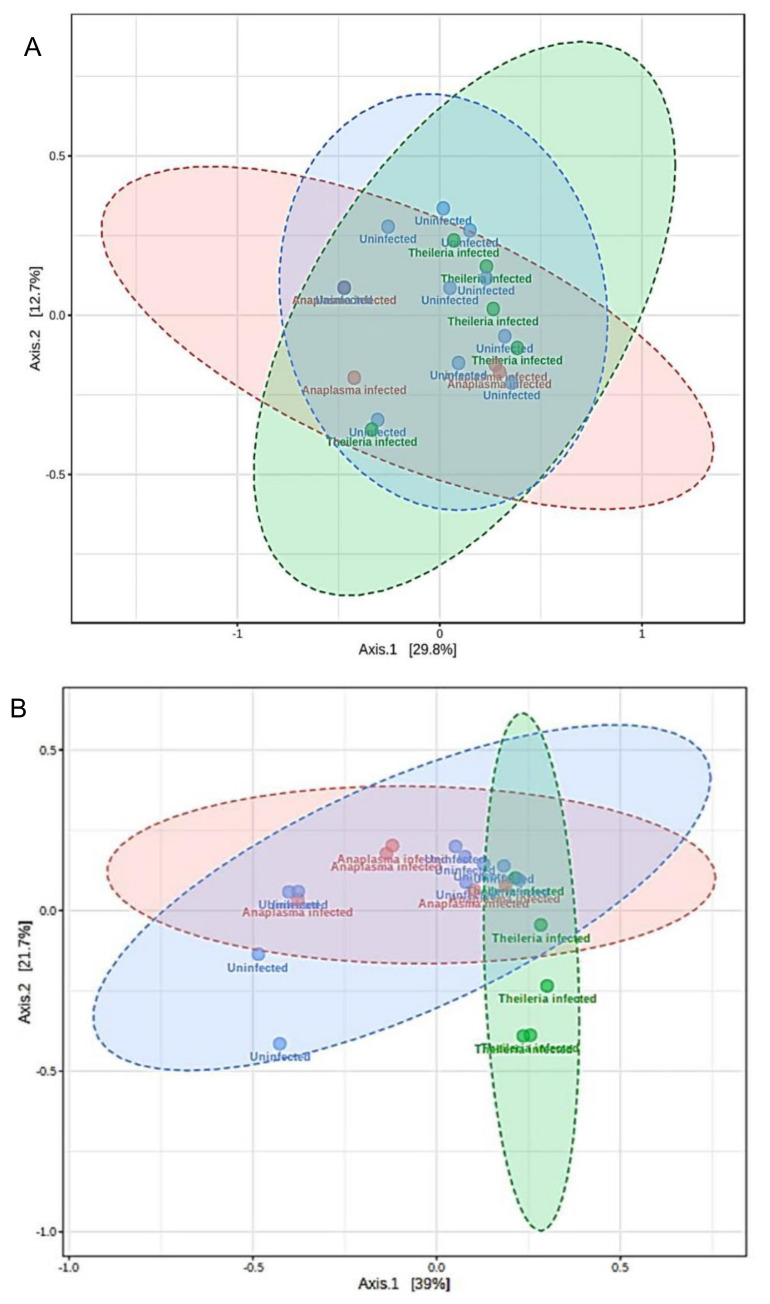
Estimation of differences in the microbial communities as a measure of Beta diversity analysis in *R. microplus* ticks. (**A**) Principal coordinate analysis (PCoA) of the Bray–Curtis distance matrix (PERMANOVA, F-value: 4.2171; R: 0.33161; *p*-value < 0.005). (**B**) Principal coordinate analysis (PCoA) Jaccard distance matrix (PERMANOVA, F-value: 3.2588; R: 0.27714; *p*-value < 0.005).

**Figure 5 microorganisms-08-01299-f005:**
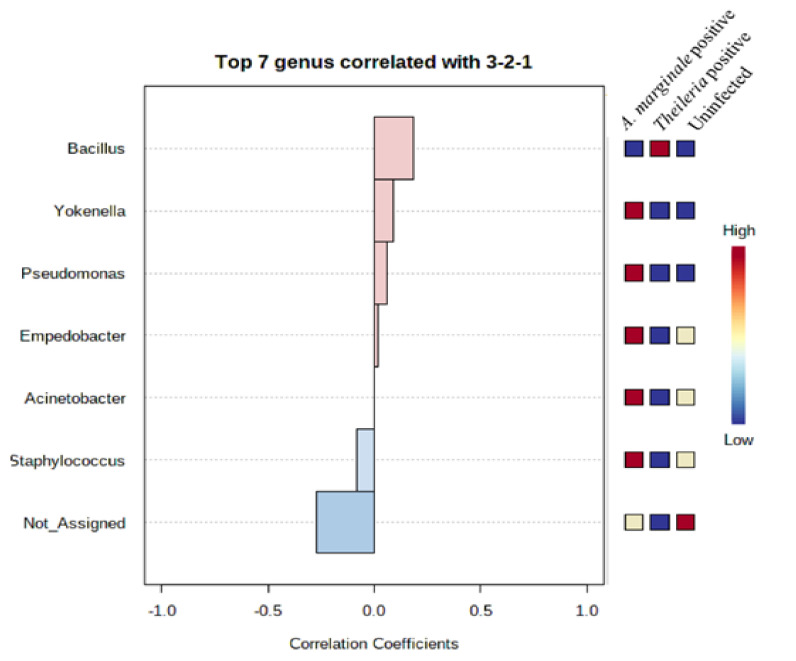
Pattern correlation analysis of top 7 bacteria genera in *R. microplus* ticks. Ticks positive with *Theileria* sp. showed a positive correlation with the presence of Bacillus.

**Figure 6 microorganisms-08-01299-f006:**
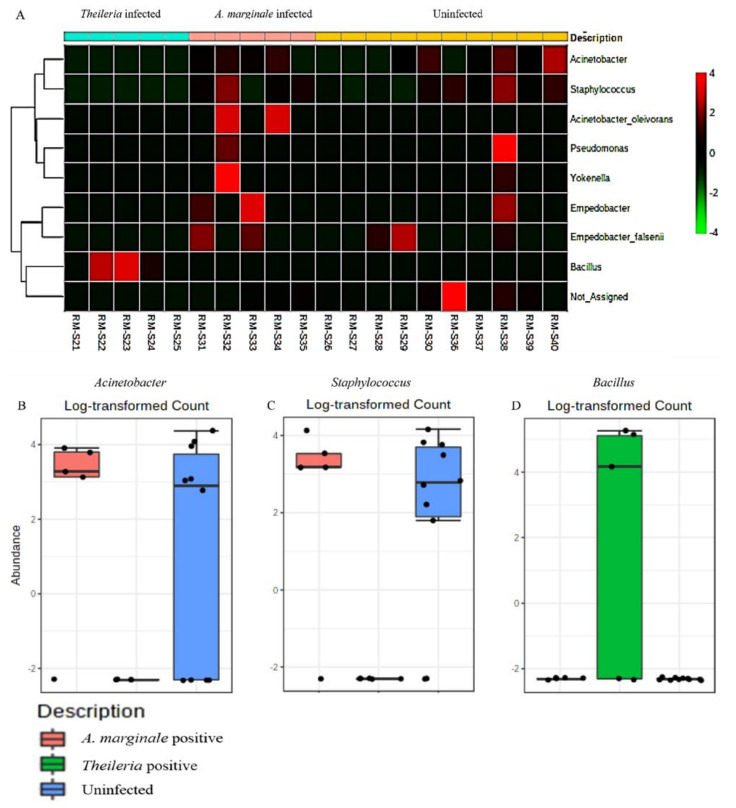
Heat map and classical univariate compositional analysis in *R. microplus* ticks. (**A**) Heat map correlation analysis between *A. marginale* positive, *Theileria* sp. positive and uninfected ticks. Log-transformed count of (**B**) Acinetobacter (FDR = 0.19277, df = 1, *p*-value = 0.007), (**C**) Staphylococcus (FDR = 0.19277, df = 1, *p*-value = 0.007), and (**D**) Bacillus (FDR = 0.0002, df = 1, *p*-value = 3.94e-223) in *A. marginale* positive, *Theileria* sp. positive and uninfected *R. microplus* ticks.

**Table 1 microorganisms-08-01299-t001:** List of primers used in this study and their respective amplicon sizes.

Target Genes	Primer Sequence (5’–3’)	Amplicon Size (bp)	References
Amar 16S-F	GGC GGT GAT CTG TAG CTG GTC TGA	270 bp	[16]
Amar 16S-R	GCC CAA TAA TTC CGA ACA ACG CTT		
*Theileria* sp 18S-F	GGT AAT TCC AGC TCCAAT AG	300 bp	[4]
*Theileria* sp 18S-R	ACC AAC AAA ATA GAA CCA AAG TC		
16S rRNA 27F	AGR GTT TGA TCM TGG CTC AG	V1–V3	[13]
16S rRNA 519R	GTN TTA CNG CGG CKG CTG		
COI-F	GGT CAA CAA ATC ATA AAG ATA TTG G	708 bp	[15]
COI-R	TAA ACT TCA GGG TGA CCA AAA AAT CA		
*Wolbachia* sp GroEL-F	TGT ATT AGA TGA TAA CGT GC	800 bp	[17]
*Wolbachia* sp GroEL-R	CCA TTT GCA GAA ATT ATT GCA		

## Data Availability

The datasets supporting the conclusion of this article are included within the article and its additional files. Raw data are available from the corresponding author upon request.

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
