# Peer review of "Tick-Borne Pathogens Shape the Native Microbiome Within Tick Vectors"

_microorganisms, 2020, doi:10.3390/microorganisms8091299_

Round 1

Reviewer 1 Report

The manuscript is good study with interesting data.  However, I have few concerns that I highlighted in the manuscript it self.

Author Response

We addressed the main comments and italicized the scientific names in the revised manuscript.

Reviewer 2 Report

The authors present a microbiome study that explores the interaction of two tick-borne pathogens with the microbiome of ticks in Pakistan. Their results indicate that Theilera sp. can manipulate the microbiome of the ticks by unknown mechanisms, leading to an increase in Bacillus sp bacteria and a decrease in bacterial diversity. Whether this is due to specific interactions between these microorganisms or the activation of the tick immune response is unknown. The ability of tick-borne pathogens to manipulate the microbiome of a tick vector has already been reported in Ixodes scapularis during infection with Anaplasma phagocytophilum. This pathogen manipulates the expression of a cold tolerance protein that also appears to be bactericidal. Interestingly, the effect of Theileria appears to be tick species-specific, as it only occurs in Rhipicephalus microplus, which indicates that it may be a pathogen-tick interaction as reported by Ixodes.

However, these results come into question after the phylogenetic analysis of the COI sequences. According to the results reported herein, the sequences of the ticks collected here are more similar to each other than to other ticks of the same species. This may be due to improper analysis (not trimming the sequences properly after the alignment, wrong portions of the gene used, etc). Nevertheless, the sequences are not provided, making it difficult for the reader to perform an analysis on their own. Whereas the 16s analysis seems sound, if the ticks are the same species, all the results come into question.

Major comments:

Figure 2: Since Plasmodium has been added to the list. The authors should not call the figure 16s bacteria abundance, but rather “microbial abundance” and eliminate the 16s.  

Figure S1: How come the out-group used is the 16s of a bacterial pathogen instead of the COI from another arthropod or a far related tick? The groupings have poor support in bootstrap. Also, you would expect that the majority of the ticks from the same species group together, but that is not the case. Rather than grouping with ticks of their same species, the COI from the samples collected in this study grouped with each other. How can the authors explain this? The submission of the sequences obtained from the ticks to GenBank is necessary. Further, additional supplementary tables are needed like those showing the percentage of identities obtained during alignment. A figure showing the alignments of a couple of sequences may also help the reader. Giving that the findings are species-specific, it is therefore required that the identity of the ticks is certain.

Another necessary piece of information would be a table showing the results from the BLAST. What are species that result from the BLAST? And what are the e-values?

The reviewer performed an NJ phylogenetic analysis of COI sequences in GenBank with COI from Culioides as the out-group and the expected grouping is obtained (see attachment).

Figure S2: The authors are misinterpreting their results. Wolbachia from Hyalomma and Rhipicephalus are closely related to each other. Their sequence appears to be almost the same. That grouping is in a different clade from wAlbB-FL2016, which separation is supported by the 63 bootstraps. Because the distances were not provided, it is hard to interpret how similar they are to other Wolbachia. This analysis may benefit from having the GroEL from Anaplasma as an out-group. As requested for the ticks, the sequences of these Wolbachia should be deposited on GenBank.

Minor comments:

  • Page 2 Line 85: Rhipicephalus microplus should be italicized.
  • Page 3 Line 111: Anaplasma marginale should also be in italics.
  • Page 4 Line 150: the article is about ticks, yet they are talking about mosquito groups.
  • Page 4 Line 165: “determined”.
  • Page 12 Line 298: “ticks” instead of “ricks”.

Author Response

Please see attached point by point rebuttal.
